# Biocompatibility of a High-Plasticity, Calcium Silicate-Based, Ready-to-Use Material

**DOI:** 10.3390/ma13214770

**Published:** 2020-10-26

**Authors:** Tomoharu Okamura, Liji Chen, Nobuhito Tsumano, Chihoko Ikeda, Satoshi Komasa, Kazuya Tominaga, Yoshiya Hashimoto

**Affiliations:** 1Department of Oral Pathology, Osaka Dental University, 8-1, Kuzuhahanazonocho, Hirakata, Osaka 573-1121, Japan; okamu-t@cc.osaka-dent.ac.jp (T.O.); ikeda-c@cc.osaka-dent.ac.jp (C.I.); tominaga@cc.osaka-dent.ac.jp (K.T.); 2Department of Orthodontics, Osaka Dental University, 8-1, Kuzuhahanazonocho, Hirakata, Osaka 573-1121, Japan; chen-li@cc.osaka-dent.ac.jp; 3Department of Oral and Maxillofacial Surgery II, Osaka Dental University, 8-1, Kuzuhahanazonocho, Hirakata, Osaka 573-1121, Japan; tsumano@cc.osaka-dent.ac.jp; 4Department of Removable Prosthodontics and Occlusion, Osaka Dental University, 8-1, Kuzuhahanazonocho, Hirakata, Osaka 573-1121, Japan; komasa-s@cc.osaka-dent.ac.jp; 5Department of Biomaterials, Osaka Dental University, 8-1, Kuzuhahanazonocho, Hirakata, Osaka 573-1121, Japan

**Keywords:** calcium-silicate, Bio-C Sealer, root-end filling material, mineral trioxide aggregate (MTA) cement, biocompatibility, beagle dog

## Abstract

The Bio-C Sealer is a recently developed high-plasticity, calcium-silicate-based, ready-to-use material. In the present study, chemical elements of the materials were characterized using scanning electron microscopy (SEM), X-ray diffraction (XRD), and Fourier transform infrared spectroscopy (FTIR). The biocompatibility of the Bio-C Sealer was investigated using cytotoxicity tests and histological responses in the roots of dogs’ teeth. XRD, SEM, and FTIR produced hydrated calcium silicate in the presence of water molecules. In addition, FTIR showed the formation of calcium hydroxide and polyethylene glycol, a dispersing agent. The 1:4 dilutions of Bio-C Sealer presented weaker cytotoxicity than the Calcipex II in an in vitro system using the V-79 cell line. After 90 d, the periradicular tissue response of beagle dog roots was histologically evaluated. Absence of periradicular inflammation was reported in 17 of the 18 roots assessed with the Bio-C Sealer, whereas mature vertical periodontal ligament fibers were observed in the apical root ends filled with the Bio-C Sealer. Based on these results and previous investigations, the Bio-C Sealer is recommended as an effective root-end filling material. These results are relevant for clinicians considering the use of Bio-C Sealer for treating their patients.

## 1. Introduction

Calcium hydroxide (Ca(OH)_2_) is an inorganic compound commonly used in dentistry; it has become one of the most widely accepted materials in dental clinics [1]. However, although permanent root canal sealers containing Ca(OH)_2_ have been introduced into the market [2], their disadvantages have also been reported, including (a) lack of adhesion to dentin and other filling materials, (b) lack of proper sealing, and (c) presence of micropores, which form tunnel defects that promote bacterial infiltration in the pulp. Therefore, more biocompatible and complex materials are now being produced to solve such issues [3]. Calcium silicate-based cement, conventionally known as mineral trioxide aggregate (MTA) cement, is biocompatible, prevents microleakage, and promotes the regeneration of original tissues when in contact with periradicular tissues [4]. In addition, it is mainly composed of dicalcium and tricalcium silicates with unique properties and a hydraulic self-curing material that makes it suitable for clinical use [5,6,7,8,9].

Therefore, a single-cone technique with a hydraulic calcium silicate-based sealer may represent a good tool given the superior flowability of the material and its ability to slightly expand upon setting [10,11]. Elizabeth et al. [10] evaluated the outcome of nonsurgical root canal treatments using a single-cone technique and a hydraulic calcium silicate-based sealer. Recently, the Bio-C Sealer (endodontic sealer, Angelus, Londrina, Paraná, Brazil) was introduced in the market in the form of a premixed bioceramic material, providing the same biological interactions as MTA, although with improved manipulation and insertion [7]. The Bio-C Sealer contains calcium silicates, which become hydrated on contact with local humidity and produce a hydrated calcium silicate structure. In turn, the formed Ca(OH)_2_ dissociates rapidly into Ca^2+^ and OH^−^ ions, thereby increasing the pH of the medium and making the environment inhospitable for bacterial growth [12]. Furthermore, the Bio-C Sealer also contains zirconium oxide as a radiopacifier [13].

Given that achieving proper root canal filling is important for the long-term success of endodontically treated teeth, root canal sealers must have appropriate physical and chemical properties [14]. Zordan-Bronzel et al. [15] evaluated the physicochemical properties of the new Bio-C Sealer in comparison with the TotalFill BC calcium silicate-based sealer and the AH Plus epoxy resin-based sealer using conventional and micro-computed tomography (CT) tests. They concluded that the Bio-C Sealer has a short setting time, alkalinization ability, and adequate flow and radiopacity.

In endodontic studies, knowing the toxicity and biocompatibility of new materials before applying them clinically is essential, as some constituent compounds can damage the surrounding tissues [16,17]. More recently, researchers evaluated the superior cytocompatibility of the Bio-C Sealer in terms of cytotoxicity in comparison with AH Plus and demonstrated that the composition of a filling material plays an important role in its biological properties [18]. However, the cytotoxicity of root canal sealers containing Ca(OH)_2_ has not been evaluated. In addition, there appears to be little histopathological information available concerning the use of a high-plasticity calcium-silicate-based material in root canals. 

Therefore, the aim of this study was to address this gap in the current literature by investigating the biocompatibility of a high-plasticity calcium-silicate-based material, the Bio-C Sealer, in the roots of dogs’ teeth.

## 2. Materials and Methods 

### 2.1. Preparation and Characterization

For materials that required moisture for setting, the mold was placed on a glass plate. Briefly, 2 g of Bio-C Sealer (Angelus, Londrina, Paraná, Brazil) was mixed according to the manufacturer’s instructions with 0.02 mL of water and the mold was filled to a slight excess. Another glass plate lined with a plastic sheet on top of the sealer was pressed and the mold was placed in the cabinet for 24 h. Thereafter, the specimens were carefully removed from the mold and the periphery of the specimen was finished to remove flash and irregularities. Successively, the crystal phase of the clinker and setting material was identified using an XRD system (XRD-6100, Shimadzu, Kyoto, Japan) and characterized using a database from the International Center for Diffraction Data. Patterns were obtained under the following conditions: 30.0 mA, 40.0 kV, scan rate: 2°/min, and 10°–70°. Scanning electron microscopy images were obtained using parameters of 5.0 kV and 10 µA (SEM; S-4100, Hitachi High-Technologies Corporation, Tokyo, Japan). Before observation, the samples were coated with platinum-palladium alloy using E-1030 (Hitachi High-Technologies Corporation, Tokyo, Japan). Then, attenuated total reflection Fourier transform infrared spectroscopy (IRAffinity-1S, Shimadzu, Kyoto, Japan) was used to evaluate the clinker and setting material over a range of 4000–400 cm^−1^ with a resolution of 2 cm^−1^.

### 2.2. Cytotoxicity Test 

This experiment involved comparisons between the Bio-C Sealer and Calcipex II (Nippon Shika Yakuhin Co., Ltd. Shimonoseki, Japan). There are very few premixed materials available for use in endodontic treatment, and we considered it reasonable to compare Bio-C Sealer and Calcipex II because both are premixed, Ca(OH)_2_-based endodontic materials. Briefly, the cytotoxic potential of these dental materials was evaluated in a 2–(4,5–dimethyl–2–thiazolyl)–3,5–diphenyl–2H–tetrazolium bromide (MTT) reduction assay using the V-79 cell line. The V-79 cell line was composed of fibroblasts isolated from the lung tissue of young Chinese male hamsters. The procedure was carried out according to the ISO guidelines 10993-12 and the ratio of material surface area to extraction vehicle volume was calculated as 1.5 cm^2^/mL. Furthermore, the extraction medium (Dulbecco’s Modified Eagle medium-high glucose plus 10% bovine fetal serum and 1% penicillin/streptomycin solution; 10,000 penicillin units/10 mg/mL streptomycin) was filtered with a 0.22 µm filter. Thereafter, the test item extract remained directly in contact with the cells for 24 h at 37 °C and 5% CO_2_ at the following concentrations: 12.5%, 25%, 50%, and 100%. The V-79 cells were maintained in a culture medium (composition the same as extraction medium) at 37 °C and 5% CO_2_ for approximately 48 h to promote cell proliferation and to achieve the desired confluency (approximately 80%). Successively, the fibroblasts from the V-79 cell line were cultured in 96-well cell culture plates (1 × 10^4^ cells per well) and incubated in the same culture medium for 24 h to facilitate cell monolayer formation. The culture medium was then replaced with a fresh medium either containing test substance (*n* = 6) or no test substance (*n* = 6). Different concentrations of the test substance used for the treatment were incubated for 24 h. At the end of the exposure period, the culture medium with the treatments was discarded and 50 µL of the MTT solution (1 mg/mL) was added to each well. Thereafter, the cells were incubated for 2 h, the MTT solution was discarded, 100 µL of isopropanol was added to each well, and the plate was shaken for 5 min. Finally, the absorbance of the test substance was measured at a wavelength of 570 nm (reference wavelength: 650 nm). This experiment was repeated at least three times. The data obtained were analyzed using Statcel, version 4 (OMS, Tokyo, Japan), and the Students’ *t*-test was applied. The level of significance was set to *p <* 0.05.

### 2.3. Animal and Tooth Preparation

Twenty teeth from the second and third maxillary premolars, as well as the second, third, and fourth mandibular premolars of four 12-month-old beagle dogs were evaluated. All animal procedures were performed according to the Animal Research Ethics Committee of the Osaka Dental University (19-02012a). Briefly, the animals were anesthetized 15 min before the operation and received inhalational anesthesia with isoflurane. Before the treatment, their teeth were radiographed and cleaned with pumice (Meiji Co. Ltd., Tokyo, Japan), while their surfaces were thoroughly wiped with a 10% povidone-iodine solution (Meiji Co. Ltd., Tokyo, Japan) to ensure aseptic conditions. The access cavity was prepared with a sterile, diamond point FG (Shofu Inc., Kyoto, Japan) under saline irrigation. In addition, the canals were prepared using Tri Auto ZX2 (J. Morita Corp., Kyoto, Japan) and apically enlarged to the working length with K-type hand files of sizes 30–50 based on the canal’s size. Furthermore, the canals were intermittently irrigated with 10 mL of 2.5% NaOCl, rinsed for 30 s with 3% EDTA (Nippon Shika Yakuhin Co., Ltd. Shimonoseki, Japan), flushed with 5 mL of sterile saline, and dried with paper points (Ci Medical Co., Ltd., Ishikawa, Japan).

Thereafter, the prepared roots were assigned to either the Bio-C Sealer or the Calcipex II group. Given that gutta-percha points were used to condense the sealers on the root canal walls, a decision was made to investigate the reactions caused by both the points and the sealers. In the Bio-C Sealer group (*n* = 36 roots), the roots were filled using the Bio-C Sealer and gutta-percha points by lateral condensation and the teeth were restored with a light-cured, glass-ionomer cement base. A similar procedure was followed for the Calcipex II group (*n* = 20 roots); however, Calcipex II was used for the filling. All the animals were euthanized with an overdose of pentobarbital after 28 days and 90 days. The specimens were removed immediately, each with one root and the surrounding alveolar bone, and placed in 10% formalin. Successively, they were demineralized in 10% formic acid, processed and embedded in paraffin, cut into approximately 4–6 µm sections, and stained with hematoxylin and eosin. Finally, the serial sections were microscopically examined and the periapical inflammation (PI) was semi-quantitatively graded according to the following scale: 0, none; 1, mild; 2, moderate; 3, severe; 4, tissue necrosis.

## 3. Results

### 3.1. Characterization of Bio-C Sealer

The results of the XRD analysis of the clinker and setting material are shown in Figure 1 and Figure 2. The XRD patterns of the clinker suggested that the mineral phase was composed largely of dicalcium silicate (C_2_S), tricalcium silicate (C_3_S), tricalcium aluminate (C_3_A), and zirconium oxide (ZrO_2_) as the contrast agent (Table 1) [19,20,21]. In contrast, the setting material showed large peaks that represented calcium silicate hydrate (CSH) (Table 2) [19,20,22]. Initially, stereomicroscopy and SEM were performed on both the clinker and the material discs (Figure 3), which showed a lump of clinker sized 300–1000 µm (Figure 3). In the SEM observation (Figure 3), the size of the calcium silicate grains in the clinker before hydration was around 2 µm. Images of the setting material indicated the formation of the CSH phase after the hydration process (average size of these grains was approximately 5–10 µm) (Figure 3). Therefore, the materials were analyzed using FTIR to determine changes between the clinker and the setting materials (Figure 4, Table 3 and Table 4). The clinker showed C_3_S and C_2_S bands of 573 cm^−1^, 1112 cm^−1^, 494 cm^−1^, and 912 cm^−1^, respectively [23]. The CSH bands were visible in the setting material at 494 cm^−1^, 575 cm^−1^, 881 cm^−1^, 1647 cm^−1^, and 3431 cm^−1^, whereas a Ca(OH)_2_ band was seen at 3637 cm^−1^ [24]. Polyethylene glycol, in other words, a dispersing agent, showed bands at 1101 cm^−1^, 1246 cm^−1^, 1348 cm^−1^, 1456 cm^−1^, 1647 cm^−1^, and 2868 cm^−1^ [25].

### 3.2. Cytotoxicity Test for the Bio-C Sealer and Calcipex II

A reduction in cell viability (below 30%) was observed for the Bio-C Sealer test substance compared to the negative control at the following concentrations: 12.5%, 25%, and 50%. The same observations were valid at a concentration of 100%. With regards to Calcipex II, a reduction in cell viability (below 30%) was observed as compared to the negative control at the 12.5% and 25% concentrations. Similar observations were made at concentrations of 50% and 100% (Figure 5).

### 3.3. Histochemical Analysis

Clinically, evidence of swelling associated with the teeth treated with the Bio-C Sealer and Calcipex II was not reported at 28 days and 90 days from the procedure. Similarly, periapical lesions were not observed following radiographic assessments (Figure 6 and Figure 7). After 28 days and 90 days, both the Bio-C Sealer- and Calcipex II-treated teeth showed immature periodontal ligament fibers and a thick layer of cementum at the apex (Figure 6b,c,e,f). In addition, infiltration of round cells (i.e., mesenchymal cells involved in tissue repair) was seen in the same area (Figure 6b,c,e,f). Although the capillaries in the surrounding tissues were hyperemic, they did not show infiltration of lymphocytes or plasma cells (Figure 6b,c,e,f). Overall, evidence of an inflammatory response associated with any of the treated teeth at 28 days was not present. At 90 days, characteristic findings of healing were seen at the apical foramen of Bio-C Sealer-treated teeth, whereas the original apical hole was reported to be closed by the growth of a bone-like hard tissue (Figure 7b). In addition, mature vertical periodontal ligament fibers were observed at the apical root portion (Figure 7c) from the alveolar bone to the apical cementum. Conversely, although mature periodontal ligament fibers were observed at the apical root portion of Calcipex II-treated teeth at 90 days (Figure 7f), they did not grow vertically.

Table 5 shows the scores of periapical inflammation (PI) assessed histologically at 28 days and 90 days after the experimental treatment procedures were completed. At 28 days and 90 days, none of the roots showed moderate or severe PI or tissue necrosis in both the Bio-C Sealer and control groups. Mild PI was reported in only 1 of the 18 roots (6%) in the Bio-C Sealer group and 2 of the 10 roots (20%) in the Calcipex II group. Such PI measures remained constant for both groups at 90 days. 

## 4. Discussion

In the present study, roots filled with the Bio-C Sealer demonstrated outstanding healing of periapical tissue, which was evaluated using histological methods. The current study is the first in vivo experiment to test this new bioceramic material.

Characterization is an important step in the search for optimal materials given that changes may occur in the substance, such as hydration of calcium silicate compounds in calcium-based materials. In the present experiment, XRD was used for both the compositional and crystal phase characterization of the Bio-C Sealer and the presence of the following mineral phases was indicated: Ca_2_SiO_5_, Ca_3_SiO_5_, and Ca_3_Al_2_O_3_ (i.e., the main constituents of Portland cement). Although the addition of ZrO_2_ provided radiopacity—a basic requirement of any dental material—the peak intensity remained high. An important point is that ZrO_2_ has much higher crystallinity and peak intensity than clinker. This fact can be explained by the amount of radiopacifier, which is much higher than the amount of clinker, and the size of the particles. Marked intensity peaks assigned to CSH in the setting materials were observed. In addition, water molecules came in contact with the Bio-C Sealer clinker, causing hydration and setting of the cement. These chemical reactions involve the hydration of calcium silicate compounds to produce a CSH gel, which is responsible for setting and forming Ca(OH)_2_ [26,27,28]. Furthermore, SEM images showed the deposition of the hydro compounds resulting from the setting reaction on the surface of the cement, as these grains grow and unite. FTIR showed the Ca(OH)_2_ band at 3640 cm^−1^ gradually sets with time and forms Ca(OH)_2_ in the setting materials.

The MTT assay is a classic test used to evaluate the possible cytotoxic effects of materials and is employed after 24 h of cell exposure as per the International Organization for Standardization (ISO) 10993 guidelines [29]. The V-79 cell line was used in the ISO 10993 guidelines and is easily maintained in culture. Furthermore, donor biopsy variability was eliminated and greater reproducibility was possible [30]. MTT assays in the present study revealed the Bio-C Sealer to have high cell viability rates at all the dilutions tested (undiluted, 1/2, 1/4), whereas strong cytotoxicity was observed with 1/2 dilutions of Calcipex II at the same time points. Calcipex II is a Ca(OH)_2_-based sealer and the cytotoxicity of such material is known to be due to the Ca(OH)_2_, which produces a high pH [31]. Given that the samples in the present study were uncured, both components might have been involved in cytotoxicity in this experiment. Nonetheless, the components might elute even in cured samples given their high solubility, resulting in cytotoxicity. Sergio López-García et al. [7] showed that the Bio-C-Sealer had higher cell viability than AH Plus containing formaldehyde and bisphenol-A, both of which are known for their toxicity. In the present study, cytotoxicity tests were conducted using cell lines. Much like target cells, primary cells may have specific metabolic potentials; therefore, they may be more consistent with in vivo conditions [32,33]. Meanwhile, cytotoxicity tests for periodontal ligament cells or dental pulp cells will be needed in the future.

The in vivo response of periradicular tissues to various root canal sealers or root filling materials has been previously investigated [12,34,35,36]. Bernabé et al. [34] demonstrated that the periradicular tissues of roots filled with a zinc oxide eugenol sealer had more severe inflammation than those filled with gray MTA. Furthermore, Torabinejad et al. [37] reported that all root-ends of monkeys that were filled with amalgam had periradicular inflammation. Supporting the present results, Chene et al. [38] showed minimal or no inflammatory reaction in the tissues adjacent to the gray MTA and proved that healing took place at all the surgical sites. In the present study, the use of the Bio-C Sealer resulted in a more effective healing process and showed no inflammation in the periapical tissue. In addition, the bone-like hard tissue was in close contact with the filled root end. 

Bio-C Sealer is the only ready-to-use cement with tricalcium aluminate, which is known to induce osteo-promotive and bone-remembering and is thought to contribute to the mineralization process of the periapical tissue [39]. In the present study, Calcipex II also allowed a satisfactory healing process and minimal or no inflammatory reactionl although it showed a PI score similar to that of the Bio-C Sealer. Economiades et al. [35] demonstrated that the inflammatory reaction of *C*alcipex II was more intense than that of MTA, probably due to its composition. In contrast, Snot et al. [36] used a Ca(OH)_2_-based sealer in the treatment of root canal fillings in dogs’ teeth and reported a less severe irritation in 30-day specimens, as well as areas of newly formed cementum. This observation was supported by their results obtained from 90-day specimens, which also supported the results of the present study.

Another important feature of MTA is its ability to promote the formation of cementum directly on the surface of the root tip, as reported in several animal experiments [37,38,40]. The formation of cementum adjacent to MTA might be a result of the reaction between calcium oxide in MTA and water or tissue fluids. This results in the formation of Ca(OH)_2_, which may stimulate hard tissue deposition [41]. In the present study, areas of newly formed cementum were observed; this observation was supported by the results obtained with 28-day specimens. After 90 days, some cementum surfaces were characterized by mature periodontal ligament fibers that mimicked Sharpey’s fibers. The source of new cementum is either derived from the remaining periodontal ligament (which grows from the sides) or from the ingrowing connective tissue of the bone [37].

## 5. Conclusions

The Bio-C Sealer presented weaker cytotoxicity than the Calcipex II sealer containing Ca(OH)_2_ in an in vitro system. In addition, the Bio-C sealer is biocompatible and safe to use in close contact with the periapical tissue. Although the induction of periradicular inflammation was not reported, the Bio-C sealer was shown to favor tissue repair. Furthermore, the sealer may contribute to the mineralization process of the periapical tissue, which demonstrates its bioactive potential. The findings of the present study are relevant for clinicians looking to explore the use of the Bio-C Sealer in clinical practice to achieve better outcomes in their patients.

## Figures and Tables

**Figure 1 materials-13-04770-f001:**
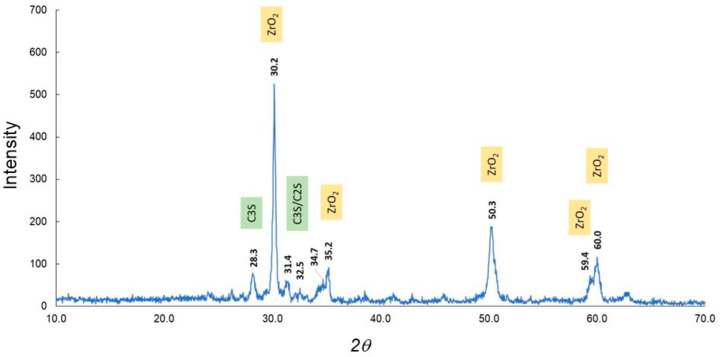
X-ray diffraction patterns for the clinker. Abbreviations: C_3_S, tricalcium silicate; C_2_S, dicalcium silicate; ZrO_2,_ zirconium oxide.

**Figure 2 materials-13-04770-f002:**
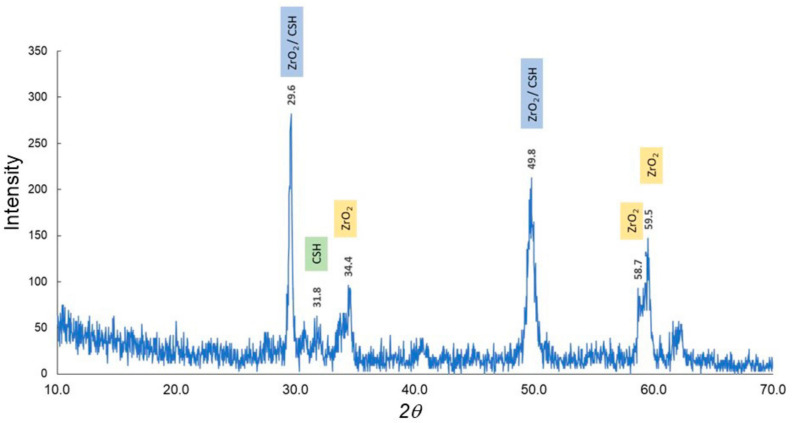
X-ray diffraction patterns for the setting materials. Abbreviations: ZrO_2,_ zirconium oxide; CSH, calcium silicate hydrate.

**Figure 3 materials-13-04770-f003:**
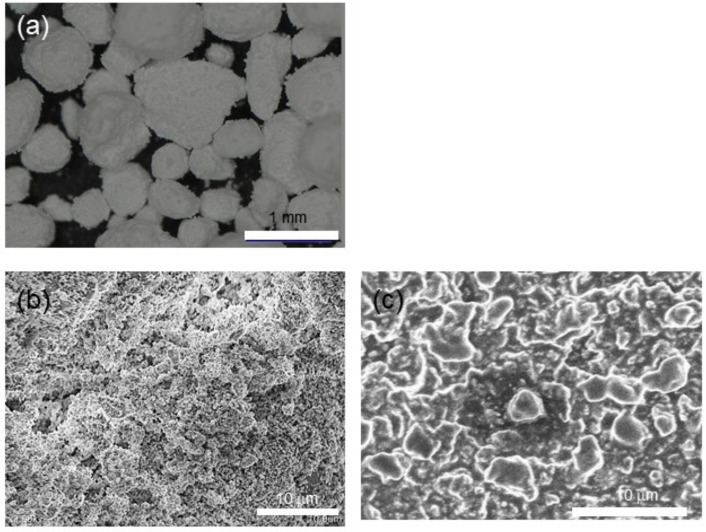
Stereomicroscope and scanning electron micrographs of the (**a**) Bio-C Sealer; (**b**) clinker; and (**c**) setting material. The bars are (**a**) 1 mm and (**b**,**c**) 10 µm.

**Figure 4 materials-13-04770-f004:**
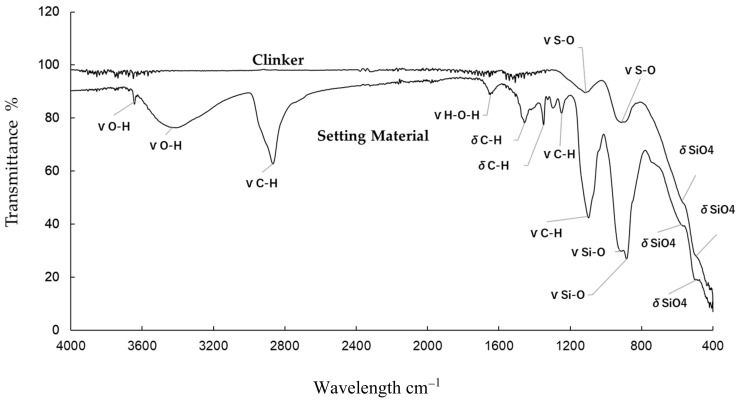
Fourier transform infrared spectroscopy (FTIR) for the clinker and setting material.

**Figure 5 materials-13-04770-f005:**
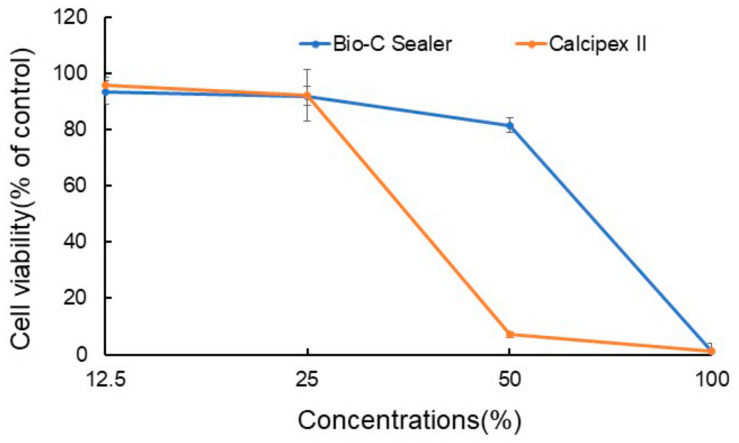
Cell viability (%) in terms of mean replicate percentages of the test substance concentrations of the Bio-C Sealer and Calcipex II.

**Figure 6 materials-13-04770-f006:**
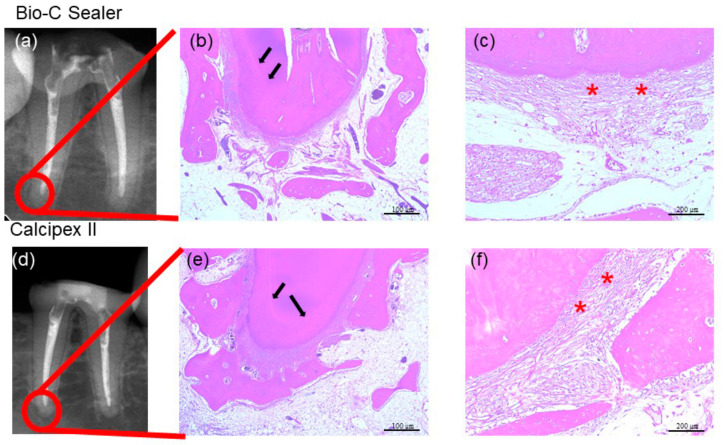
Histopathological events were observed after the evaluation of the periapical tissue response to the proposed endodontic treatments. (**a**) Radiographic image and (**b**) histological section of a Bio-C Sealer-treated tooth after 28 days; (**c**) highest magnification of the same section. (**d**) Radiographic image and (**e**) histological section of a Calcipex II-treated tooth after 28 days; (**f**) highest magnification of the same section. The bars of (**b**,**e**) are 100 µm, whereas those of (**c**,**f**) are 200 µm. Black arrows indicate cementum, whereas red asterisks indicate immature periodontal ligament fibers.

**Figure 7 materials-13-04770-f007:**
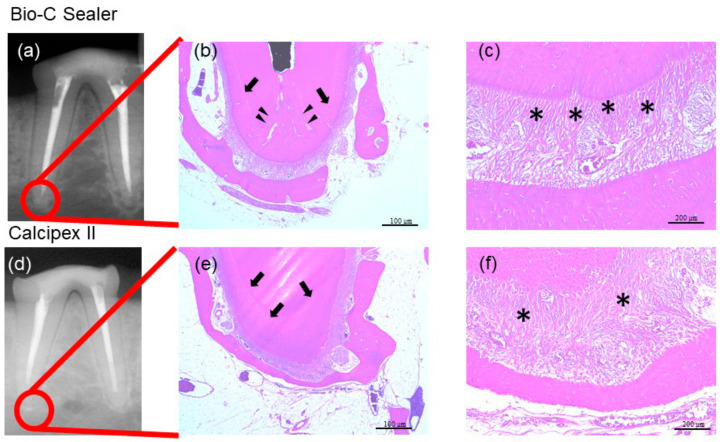
Histopathological events were observed after the evaluation of the periapical tissue response to the proposed endodontic treatments. (**a**) Radiographic image and (**b**) histological section of a Bio-C Sealer-treated tooth after 90 days; and (**c**) highest magnification of the same section. (**d**) Radiographic image and (**e**) histological section of a Calcipex II-treated tooth after 90 days; (**f**) highest magnification of the same section. The bars of (**b**,**e**) are 100 µm, whereas those of (**c**,**f**) are 200 µm. Black arrows indicate cementum, red asterisks indicate mature periodontal ligament fibers, and black triangles represent bone-like hard tissue.

**Table 1 materials-13-04770-t001:** X-ray diffraction information for the clinker.

Material	2 *θ*	Intensity	Crystal System	Crystal Planes
C_3_S	28.3	74	Hexagonal	2, 0, −2, 0
ZrO_2_	30.2	525	Tetragonal	1, 0, 1
C_3_S/ C_2_S	31.4	59	Hexagonal Hexagonal	0, 0, 0, 2 1, 0, −1, 2
C_3_S/ C_2_S	32.5	43	Hexagonal Hexagonal	2, 0, −2, 1 2, −1, −1, 0
ZrO_2_	35.2	90	Tetragonal	1, 1, 0
ZrO_2_	50.3	187	Tetragonal	1, 1, 2
ZrO_2_	59.4	72	Tetragonal	1, 0, 3
ZrO_2_	60.0	116	Tetragonal	2, 1, 1

Abbreviations: C_3_S, tricalcium silicate; C_2_S, dicalcium silicate; ZrO_2,_ zirconium oxide.

**Table 2 materials-13-04770-t002:** X-ray diffraction information for the setting materials.

Material	2 *θ*	Intensity	Crystal System	Crystal Planes
ZrO_2_/CSH	29.6	282	Tetragonal Orthorhombic	1, 0, 1 0, 2, 0 −2, 2, 0
CSH	31.8	63	Orthorhombic	2, −2, 2 0, 2, 2
ZrO_2_	34.4	90	Tetragonal	1, 1, 0
ZrO_2_/CSH	49.8	213	Tetragonal Orthorhombic	1, 1, 2 2, 4, 0
ZrO_2_	58.7	72	Tetragonal	1, 0, 3
ZrO_2_	59.6	147	Tetragonal	2, 1, 1

Abbreviations: ZrO_2_, zirconium oxide, CSH, calcium silicate hydrate.

**Table 3 materials-13-04770-t003:** Fourier transform infrared spectroscopy (FTIR) information for the clinker.

Material	Vibrational Mode	Wavenumber (cm^−1^) Experimental	Wavenumber (cm^−1^) (Reference)
C_3_S	vSi–O stretching vibrations	1112	∼1060 [23]
C_2_S	vSi–O stretching vibrations	912	∼925 [23]
C_3_S	δSiO_4_ bending	573	∼522 [23]
C_2_S	δSiO_4_ bending	494	∼452 [23]

Abbreviations: C_3_S, tricalcium silicate; C_2_S, dicalcium silicate.

**Table 4 materials-13-04770-t004:** Fourier transform infrared spectroscopy (FTIR) information for the setting materials.

Material	Vibrational Mode	Wavenumber (cm^−1^) Experimental	Wavenumber (cm^−1^) (Reference)
Ca(OH)_2_	vO–H stretching (portlandite)	3637	∼3644 [24]
CSH	vO–H stretching	3431	3300 to 3600 [24]
PEG	vC–H stretching mode	2868	∼2878 [25]
CHS	vH–O–H stretching vibrations	1647	∼1640 [24]
PEG	δC–H bending	1456	∼1464 [25]
PEG	δC–H bending	1348	∼1343 [25]
PEG	vC–O stretching vibrations	1246	1000 to 1300 [25]
PEG	vC–O stretching vibrations	1101	1000 to 1300 [25]
CSH	vSi–O stretching vibrations	925	∼1060 [24]
CSH	vSi–O stretching vibrations	881	∼900 [24]
CSH	δSiO_4_ bending	577	∼400 to 500 [24]
CSH	δSiO_4_ bending	494	∼400 to 500 [24]

Abbreviations: CSH, calcium silicate hydrate; PEG, polyethylene glycol.

**Table 5 materials-13-04770-t005:** Scores of periapical inflammations in dogs’ premolars following histological assessment at 28 days and 90 days from the experimental treatment procedures.

Groups	Roots Assessed (*n*)	Level of PI *				
		None (%)	Mild (%)	Moderate	Severe	Tissue Necrosis
Bio-C Sealer Group 28 days	18	17 (94)	1 (6)	0	0	0
Control group (Calcipex II) 28 days	10	8 (80)	2 (20)	0	0	0
Bio-C Sealer Group 90 days	18	17 (94)	1 (6)	0	0	0
Control Group (Calcipex II) 90 days	10	8 (80)	2 (20)	0	0	0

* PI, periapical inflammation.

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
