# Peer review of "Biocompatibility of a High-Plasticity, Calcium Silicate-Based, Ready-to-Use Material"

_materials, 2020, doi:10.3390/ma13214770_

Round 1

Reviewer 1 Report

In the present study, the biocompatibility of two different root sealers was compared in vitro and in vivo. In vitro, the effects of eluates of two sealer materials were tested using the V-79 cell line; in vivo, the dog teeth roots filled with two different materials were compared histologically. The data presented are new, and the study is well performed. Some points still need to be considered.

  1. The protocol for the cytotoxicity test in vitro should be provided in more detail. Such aspects as cell seeding density, medium type, medium volume, use of antibiotics, number of replicates should be provided. 
  2. The statistic section is missing. 
  3. Why were V-79 cells chosen for the cytotoxicity test? This is a cell line, and therefore it is less susceptible to the different stimuli. Experiments with primary cells would strengthen the manuscript; otherwise, it should be discussed as the limitation.
  4. The number of animals used and the number and types of teeth used for filling should be provided.

Author Response

Comments and Suggestions for Authors

In the present study, the biocompatibility of two different root sealers was compared in vitro and in vivo. In vitro, the effects of eluates of two sealer materials were tested using the V-79 cell line; in vivo, the dog teeth roots filled with two different materials were compared histologically. The data presented are new, and the study is well performed. Some points still need to be considered.

The protocol for the cytotoxicity test in vitro should be provided in more detail. Such aspects as cell seeding density, medium type, medium volume, use of antibiotics, number of replicates should be provided.

Answer:

Thank you for your careful review and valuable advice. We have updated our article to include the detailed protocol for the cytotoxicity test in vitro

The statistic section is missing.

Answer:

Thank you for pointing this out. We have added the statistical analysis at the end of the cytotoxicity study.

Why were V-79 cells chosen for the cytotoxicity test? This is a cell line, and therefore it is less susceptible to the different stimuli. Experiments with primary cells would strengthen the manuscript; otherwise, it should be discussed as the limitation.

Answer:

Thank you for the question. We used the V-79 cell line in accordance with the ISO 10993 guidelines and easily maintained it in the culture medium. Furthermore, we eliminated the donor biopsy variability to obtain greater reproducibility. However, because primary cells may have a specific metabolic potential that is similar to that of the target cells, they may be better suited for the in vivo situation. We have outlined the advantages and disadvantages of using primary cells in the text.

The number of animals used and the number and types of teeth used for filling should be provided.

Answer:

Thank you for your advice. We have provided details regarding the number of animals and the number and types of teeth used for filling in lines 121-122 on page 3.

Reviewer 2 Report

Biocompatibility of a high-plasticity calcium silicate-2 based substance as a root-end filling material

The calcium-silicate-based Bio-C Sealer was evaluated in this study. The goal is to investigate the biocompatibility of Bio-C Sealer using cytotoxicity tests and histological responses in the roots of dogs' teeth (peri-radicular tissue response of beagle dog roots).

  1. Please provide the peak locations (2θ) and crystal planes of the following crystals shown in the X-ray diffraction (XRD) patterns: dicalcium silicate, tricalcium silicate, tricalcium aluminate, zirconium oxide, and CSH as well.
  2. It should explicitly include references for FTIR absorption peaks for readers’ reference and as the credential.
  3. Authors claim that “FTIR showed that the calcium hydroxide band at 3640cm-1 gradually set with time and formed calcium hydroxide in the setting materials. The formed calcium hydroxide is known to dissociate rapidly into Ca2+ and OH- ions, thereby increasing the pH of the medium and making the environment inhospitable for bacterial growth [18]”. Is there any missing information or data to support this claim? There are no direct time variations on the formation or dissociation of calcium hydroxide provided in the manuscript.
  4. It’s a little confusing that in Figure 4 Cell viability (%) against the concentrations of Bio-C Sealer and Calcipex II.
    • If both are low then the cytotoxicity is surely low.
    • When both are high (100%), cytotoxicity is obvious.
    • Following this trend, Bio-C Sealer is only better off than Calcipex II if the concentration of use is higher than 25% up to, say, slightly higher than 50% or so. What’s the standard (or recommended) dosage of Calcipex II? Can Bio-C Sealer take the advantage over Calcipex II if practically a higher concentration is needed?
  1. What is the concentration (or any reference dosages) being introduced for the periapical inflammation (PI) at 28 and 90 days after experimental treatment procedures in the premolars of beagle dogs?
  2. Since Bio-C Sealer was put into the market recently, it should be subjected to cytotoxicity tests rigorously before. The benefits of using Bio-C Sealer is mainly due to its tricalcium aluminate content, which is osteo-promotive. So I wonder why the authors would like to re-assure the cytotoxicity of Bio-C Sealer? Authors may need to clarify their purposes at the beginning of the introduction. I try to avoid the perception that this study sounds like a promotion of the use of Bio-C Sealer.

Author Response

The calcium-silicate-based Bio-C Sealer was evaluated in this study. The goal is to investigate the biocompatibility of Bio-C Sealer using cytotoxicity tests and histological responses in the roots of dogs' teeth (peri-radicular tissue response of beagle dog roots).

Please provide the peak locations (2θ) and crystal planes of the following crystals shown in the X-ray diffraction (XRD) patterns: dicalcium silicate, tricalcium silicate, tricalcium aluminate, zirconium oxide, and CSH as well.

Answer:

Thank you for your input. We have attempted to clarify the results accordingly and have added the peak value. We reviewed all the peak locations (2θ); for dicalcium silicate, tricalcium silicate, tricalcium aluminate, and zirconium oxide, we used the following databases and cited them in the text:

Jain, A.; Ong, S.P.; Hautier, G.; Chen, W.; Richards, W.D.; Dacek, S.; Cholia, S.; Gunter, D.; Skinner, D.; Ceder, G. Commentary: The Materials Project: A materials genome approach to accelerating materials innovation. APL. Mater. 2013, 1, 011002.

doi: 10.1063/1.4812323

Maddalena, R.; Li, K.; Chater, P.A.; Michalik, S.; Hamilton, A. Direct synthesis of a solid calcium-silicate-hydrate (CSH). Constr. Build. Mater. 2019, 223, 554-565

doi:10.1063/1.4812323

It should explicitly include references for FTIR absorption peaks for readers’ reference and as the credential.

Answer:

Thank you for your suggestion. We have included appropriate references for FTIR absorption peaks.

Authors claim that “FTIR showed that the calcium hydroxide band at 3640cm-1 gradually set with time and formed calcium hydroxide in the setting materials. The formed calcium hydroxide is known to dissociate rapidly into Ca2+ and OH- ions, thereby increasing the pH of the medium and making the environment inhospitable for bacterial growth [18]”. Is there any missing information or data to support this claim? There are no direct time variations on the formation or dissociation of calcium hydroxide provided in the manuscript.

Answer:

Thank you for your detailed assessment. We agree with your view, given that we did not measure the pH or check the presence of ions before and after hydration. Therefore, we believed that it would be appropriate to remove this reference from the discussion section; we suggest that the text that mentions the dissociation of ions be moved to the introduction section (page 2, lines 46-49). We seek your opinion on the same.

It’s a little confusing that in Figure 4 Cell viability (%) against the concentrations of Bio-C Sealer and Calcipex II. If both are low then the cytotoxicity is surely low. When both are high (100%), cytotoxicity is obvious. Following this trend, Bio-C Sealer is only better off than Calcipex II if the concentration of use is higher than 25% up to, say, slightly higher than 50% or so. What’s the standard (or recommended) dosage of Calcipex II? Can Bio-C Sealer take the advantage over Calcipex II if practically a higher concentration is needed?

Answer:

Thank you for your detailed question. In both clinical practice and animal studies, root canals are directly filled with either the Bio-C Sealer or Calcipex II. Therefore, we conducted our experiment according to the ISO guidelines 10993-12 to compare the cytotoxicity between materials. 

What is the concentration (or any reference dosages) being introduced for the periapical inflammation (PI) at 28 and 90 days after experimental treatment procedures in the premolars of beagle dogs?

Answer:

The dosages of the materials that cause PI are not relevant to our study. However, we have researched and described some dental materials that have been used to treat severe periapical inflammation on page 9, lines 275-284.

Since Bio-C Sealer was put into the market recently, it should be subjected to cytotoxicity tests rigorously before. The benefits of using Bio-C Sealer is mainly due to its tricalcium aluminate content, which is osteo-promotive. So I wonder why the authors would like to re-assure the cytotoxicity of Bio-C Sealer? Authors may need to clarify their purposes at the beginning of the introduction. I try to avoid the perception that this study sounds like a promotion of the use of Bio-C Sealer.

Answer:

We understand your reservations.  Calcium hydroxide is an inorganic compound commonly used in dentistry; it has become one of the most widely accepted materials in dental clinics. However, although permanent root canal sealers containing calcium hydroxide have introduced on the market. More recently, researchers evaluated the superior cytocompatibility of the Bio-C Sealer in terms of cytotoxicity in comparison with AH Plus and demonstrated that the composition of a filling material plays an important role in its biological properties. However, the comparison of cytotoxicity with root canal sealers containing calcium hydroxide is yet to be carried. We have described the rationale for conducting our cytotoxicity evaluation on introduction section.

Reviewer 3 Report

Dear Author, 

your work speaks about a new calcium-silicate-based, ready-for-use material, Bio-C Sealer with a high plasticity. Where you performed cytotoxicity and histological analysis on dogs teeth. It is an interesting paper however it has different flows and should be improved. 

Abstract: please improve this section as different materials information are missing. Please see the author guidelines of Materials journal and correct it. It is very messy and not clear at a glance for the readers. 

introduction: please better describe the studies available in the literature regarding the high- plasticity calcium-silicate-based, ready-for-use material, Bio-C Sealer and discuss why the necessity of your cytotoxicity evaluation being it almost studies by López-García et al. on Materials 2019. Thus the introduction section should be better organized. Moreover, an extensive english language revision of the whole text should be performed. 

materials and methods section: 

line 64 please in the subheading do not report the information regarding the material but report it on the text. the same for all the materials and facilities/equipments used in the study. please specify also the V-79 cell line with all the necessary information.

line 121: please specify what is PI and refer the whole name the first time you mention the word in the text then put the abbreviations. 

Results

The clinker results are not too clear in the way reported. Please put them in a table to better understand them. line 143 please refer figures like (Figure 6-b) and not as you use to report them. 

Discussion: 

please provide more studies regarding this materials that have been performed in the literature and discuss them on this section. It should be more fluently and readable. Then discuss your results in base of them. 

Conclusions: 

please do not provide references in this section. Being your study an in vitro and in vivo one in dogs, report them in base of results obtained.  

Please revise all the manuscript as different flows and typos are present. Moreover, a native english speaker should read the whole paper. 

Author Response

your work speaks about a new calcium-silicate-based, ready-for-use material, Bio-C Sealer with a high plasticity. Where you performed cytotoxicity and histological analysis on dogs teeth. It is an interesting paper however it has different flows and should be improved.

Abstract: please improve this section as different materials information are missing. Please see the author guidelines of Materials journal and correct it. It is very messy and not clear at a glance for the readers.

Answer:

Thank you for your careful review and valuable suggestions. We have corrected the abstract according to the author guidelines of Materials

introduction: please better describe the studies available in the literature regarding the high- plasticity calcium-silicate-based, ready-for-use material, Bio-C Sealer and discuss why the necessity of your cytotoxicity evaluation being it almost studies by López-García et al. on Materials 2019. Thus the introduction section should be better organized. Moreover, an extensive english language revision of the whole text should be performed.

Answer:

Thank you for your input. We have described the rationale for using our cytotoxicity evaluation on page 2, lines 63-70. The manuscript has been edited and checked by professional English language editors at Editage, a division of Cactus Communications.

Materials and methods section:

line 64 please in the subheading do not report the information regarding the material but report it on the text. The same for all the materials and facilities/equipments used in the study. Please specify also the V-79 cell line with all the necessary information.

Answer:

Thank you for your advice. We have revised line 64 and specified the details of the V-79 cell line in lines 96-97 on page 3.

line 121: please specify what is PI and refer the whole name the first time you mention the word in the text then put the abbreviations.

Answer:

Thank you for pointing this out. The term ‘PI’ in line 121 (now line 146) is the abbreviation of “periapical inflammation”, as we have stated. 

Results

The clinker results are not too clear in the way reported. Please put them in a table to better understand them.

Answer:

Thank you for your input. We have attempted to clarify the results accordingly and have added the peak value. We reviewed all the peak locations (2θ); for dicalcium silicate, tricalcium silicate, tricalcium aluminate, and zirconium oxide, we used the following databases and cited them in the text:

Jain, A.; Ong, S.P.; Hautier, G.; Chen, W.; Richards, W.D.; Dacek, S.; Cholia, S.; Gunter, D.; Skinner, D.; Ceder, G. Commentary: The Materials Project: A materials genome approach to accelerating materials innovation. APL. Mater. 2013, 1, 011002.
doi: 10.1063/1.4812323

Maddalena, R.; Li, K.; Chater, P.A.; Michalik, S.; Hamilton, A. Direct synthesis of a solid calcium-silicate-hydrate (CSH). Constr. Build. Mater. 2019, 223, 554-565

doi:10.1063/1.4812323

line 143 please refer figures like (Figure 6-b) and not as you use to report them.

Answer:

Thank you for this input. We have revised the format of the figure citation as (Figure 6-b).

Discussion:

please provide more studies regarding this materials that have been performed in the literature and discuss them on this section. It should be more fluently and readable. Then discuss your results in base of them.

Answer:

Thank you for your advice. As per your suggestion, we have revised this section as far as possible. We would appreciate your inputs on our revision.

Conclusions:

please do not provide references in this section. Being your study an in vitro and in vivo one in dogs, report them in base of results obtained. 

Answer:

Thank you for pointing this out. We have removed the references from this section and have reiterated the in vitro results.

Please revise all the manuscript as different flows and typos are present. Moreover, a native english speaker should read the whole paper.

Answer:

Thank you for the input. This manuscript was edited by professional English language editors at Editage, a division of Cactus Communications.

Reviewer 4 Report

Dear Authors , 

many thanks for submitting your work.

Please find my comments below:

INTRODUCTION

There is some confusion in regards to terminology and materials reported

Line 34- Calcium hydroxide has long been the gold standard among root filling materials. - actually CaOH is consider more a medicament rather than a permanent root filling 

Line 38 - Calcium silicate based cement - eg MTA , you should quote more relevant paper in regards to this materials such us Torabinejad et all.

When talking about bio-ceramic you should quote relevant papers such us Donnermeyer 2019 and give to the reader more information on how calcium silicate  sealers must be use ( single cone technique) and which clinical evidence it is currently available please quote Zavattini 2020, Chybowsky 2019

Materials and Methods

Despite you conducted and extensive study in animals you have wrongly compared the BioC sealer which is a premixed bio ceramic with Calcipex II which is a dressing material not to be used as a permanent root filling materials. It does not really make sense to compare the effect on cells due to the big differences in principles.

In addition the sample size is very low and statistical method are of low significance.

Results, conclusion and discussion

The wording and English style need to be completely changed if it is intended to be submitted for scientific purposes.

Author Response

Dear Authors ,

many thanks for submitting your work.

Please find my comments below:

INTRODUCTION

There is some confusion in regards to terminology and materials reported

Line 34- Calcium hydroxide has long been the gold standard among root filling materials. - actually CaOH is consider more a medicament rather than a permanent root filling

Answer:

Thank you for your careful review of this terminology. We agree with your view that CaOH is more appropriate as a medicament than a permanent root filling. We have revised the sentence in our manuscript (page 1, line 34).

Line 38 - Calcium silicate based cement - eg MTA , you should quote more relevant paper in regards to this materials such us Torabinejad et all.

Answer:

Thank you for your suggestion. We have cited the following paper:

Torabinejad, M.; Chivian, N. Clinical applications of mineral trioxide aggregate. J. Endod. 1999, 25, 197-205.

When talking about bio-ceramic you should quote relevant papers such us Donnermeyer 2019 and give to the reader more information on how calcium silicate  sealers must be use ( single cone technique) and which clinical evidence it is currently available please quote Zavattini 2020, Chybowsky 2019

Answer:

Thank you for the suggestion. We have cited a paper by Donnermeyer et al. (2019) with accompanying details on the appropriate use of calcium silicate sealers (single-cone technique) as per currently available clinical evidence (Giovarruscio et al., 2020, Chybowski et al., 2019) in pages 1-2, lines 42-47.

Materials and Methods

Despite you conducted and extensive study in animals you have wrongly compared the BioC sealer which is a premixed bio ceramic with Calcipex II which is a dressing material not to be used as a permanent root filling materials.

Answer: Thank you for your input. We have changed the title of our manuscript.

It does not really make sense to compare the effect on cells due to the big differences in principles.

Answer: Thank you for your input. We have described the rationale for using our cytotoxicity evaluation on page 2, lines 63-70.

In addition the sample size is very low and statistical method are of low significance.

Answer:

Thank you for pointing this out. We have described the statistical analysis at the end of the cytotoxicity study.

Results, conclusion and discussion

The wording and English style need to be completely changed if it is intended to be submitted for scientific purposes.

Answer:

This manuscript has been edited by professional English language editors at Editage, a division of Cactus Communications.

Round 2

Reviewer 1 Report

The authors carefully considered all raised points.

Author Response

Thank you for the careful review and valuable advice.

Reviewer 2 Report

Biocompatibility of a high-plasticity calcium silicate-2 based substance as a root-end filling material

It looks that authors don’t have a good understanding about the XRD and FTIR. Both are important tools to qualitatively determine the microstructure.

  • Please provide the “crystal planes” in each peak in XRD. Authors should know what XRD is measuring for. The “crystal phase” is not an XRD term.
  • Similarly, authors should provide the vibrational mode in each absorption peak in FTIR.

In fact, XRD and FTIR should be cross-examined to make sure what exact microstructure present in the materials. Considering the background of the authors, I didn’t ask for this cross-examination. However, simply input basic information in the report is necessary as this is the purpose of using experimental equipment. It also provides credible information for the research community.

4. It’s a little confusing that in Figure 4 Cell viability (%) against the concentrations of Bio-C Sealer and Calcipex II.

  • If both are low then the cytotoxicity is surely low.
  • When both are high (100%), cytotoxicity is obvious.

Following this trend, Bio-C Sealer is only better off than Calcipex II if the concentration of use is higher than 25% up to, say, slightly higher than 50% or so. What’s the standard (or recommended) dosage of Calcipex II? Can Bio-C Sealer take the advantage over Calcipex II if practically a higher concentration is needed?

Authors’ reply:

Thank you for your detailed question. In both clinical practice and animal studies, root canals are directly filled with either the Bio-C Sealer or Calcipex II. Therefore, we conducted our experiment according to the ISO guidelines 10993-12 to compare the cytotoxicity between materials.

I did not question the experimental procedure; my question is asking a scientific-based explanation for the advantage of Bio-C Sealer over that of Calcipex II in a certain window of concentration. It doesn’t need to be complicated but such an explanation is crucial for a continuous study on the comparison between the two agents.

Author Response

Biocompatibility of a high-plasticity calcium silicate-2 based substance as a root-end filling material

It looks that authors don’t have a good understanding about the XRD and FTIR. Both are important tools to qualitatively determine the microstructure.

Please provide the “crystal planes” in each peak in XRD. Authors should know what XRD is measuring for. The “crystal phase” is not an XRD term.

Similarly, authors should provide the vibrational mode in each absorption peak in FTIR.

In fact, XRD and FTIR should be cross-examined to make sure what exact microstructure present in the materials. Considering the background of the authors, I didn’t ask for this cross-examination. However, simply input basic information in the report is necessary as this is the purpose of using experimental equipment. It also provides credible information for the research community.

Reply: Thank you for the careful review and valuable advice. We have provided the “crystal planes” in each peak in XRD and the vibrational mode in each absorption peak in FTIR.

  1. It’s a little confusing that in Figure 4 Cell viability (%) against the concentrations of Bio-C Sealer and Calcipex II.

If both are low then the cytotoxicity is surely low.

When both are high (100%), cytotoxicity is obvious.

Following this trend, Bio-C Sealer is only better off than Calcipex II if the concentration of use is higher than 25% up to, say, slightly higher than 50% or so. What’s the standard (or recommended) dosage of Calcipex II? Can Bio-C Sealer take the advantage over Calcipex II if practically a higher concentration is needed?

Authors’ reply:

Thank you for your detailed question. In both clinical practice and animal studies, root canals are directly filled with either the Bio-C Sealer or Calcipex II. Therefore, we conducted our experiment according to the ISO guidelines 10993-12 to compare the cytotoxicity between materials.

I did not question the experimental procedure; my question is asking a scientific-based explanation for the advantage of Bio-C Sealer over that of Calcipex II in a certain window of concentration. It doesn’t need to be complicated but such an explanation is crucial for a continuous study on the comparison between the two agents.

Reply: As described in the discussion part, given that the samples in the present study were uncured, both components might have been involved in cytotoxicity in this experiment. Nonetheless, the components might elute even in cured samples given their high solubility, resulting in cytotoxicity. It is considered that Bio-C Sealer gradually hardens in the culture medium, so that the elution of components decreases, and as a result, the cytotoxicity is weaker than that of Calcipex II. In this study, the amount of components eluted in the culture medium was not measured.

Reviewer 3 Report

Dear authors, 

you improved you paper in base of the comments that the reviewer did. However, there are still some flows and english mistakes in the whole text. Please, follow carefully the author guidelines of the materials journal before revise the manuscript. In the abstract do not write the subheads like: 1) background, 2) materials and methods etc. Please refresh the references section also. Eliminate the not mandatory references and replace them with up dated one.  The whole text should be revised by a native english speaker. 

Author Response

you improved you paper in base of the comments that the reviewer did. However, there are still some flows and english mistakes in the whole text. Please, follow carefully the author guidelines of the materials journal before revise the manuscript. In the abstract do not write the subheads like: 1) background, 2) materials and methods etc. Please refresh the references section also. Eliminate the not mandatory references and replace them with up dated one.  The whole text should be revised by a native english speaker.

Reply: Thank you for your careful review and valuable advices. The subheadings were deleted and the references section was revised. The whole text was checked and edited by professional English language editors at Editage, a division of Cactus Communications.

Reviewer 4 Report

Dear Authors, 

thanks for reviewing your manuscript.

Despite the effort I'm not sure if this can be an acceptable work to be published

because of the following:

  • even you have changed the title you are still comparing two materials with different purposes and this does not make any sense in clinical dentistry.
  • The English is not well reviewed and instead of using the wording " we performed this experiment......we evaluated......." you should write in third person such us " the experiment was performed.......the evaluation consisted in......etc etc

Author Response

Dear Authors,

thanks for reviewing your manuscript.

Despite the effort I'm not sure if this can be an acceptable work to be published

because of the following:

even you have changed the title you are still comparing two materials with different purposes and this does not make any sense in clinical dentistry.

Reply: Thank you for your careful review and valuable advices.The scientific rationale behind this study is to compare the cytotoxicity of Calcipex II, which is mainly composed of calcium hydroxide, and Bio-C Sealer, which produces calcium hydroxide secondarily. In this version, We have provided the “crystal planes” in each peak in XRD and the vibrational mode in each absorption peak in FTIR.

The English is not well reviewed and instead of using the wording " we performed this experiment......we evaluated......." you should write in third person such us " the experiment was performed.......the evaluation consisted in......etc etc

Answer; The whole text was edited by professional English language editors at Editage, a division of Cactus Communications.